# Joint COVID-19 Contact Tracing and Malaria Reactive Case Detection as Efficient Strategies for Disease Control

Ebenezer Krampah Aidoo [1,*], Daniel Sai Squire [2], Obed Ohene-Djan Atuahene [3], Kingsley Badu [4], Felix Abekah Botchway [1], George Osei-Adjei [1], Samuel Asamoah Sakyi [5], Linda Amoah [6], Michael Appiah [1], Ruth Duku-Takyi [1], Richard Harry Asmah [7], Bernard Walter Lawson [4] and Karen Angeliki Krogfelt [8,9]

[1] Department of Medical Laboratory Technology, Accra Technical University, Accra GP 561, Ghana
[2] Department of Medical Laboratory Sciences, School of Allied Health Sciences, University of Health & Allied Sciences, Ho PMB 31, Ghana
[3] Department of Basic Medical Sciences, School of Medicine, University of Health & Allied Sciences, Ho PMB 31, Ghana
[4] Department of Theoretical & Applied Biology, Kwame Nkrumah University of Science & Technology, University Post Office, Kumasi AK 039, Ghana
[5] Department of Molecular Medicine, Kwame Nkrumah University of Science & Technology, University Post Office, Kumasi AK 039, Ghana
[6] Department of Immunology, Noguchi Memorial Institute for Medical Research, University of Ghana, Accra LG 581, Ghana
[7] Department of Biomedical Sciences, School of Basic and Biomedical Science, University of Health & Allied Sciences, Ho PMB 31, Ghana
[8] Department of Science and Environment, Unit of Molecular and Medical Biology, The PandemiX Center, Roskilde University, 4000 Roskilde, Denmark
[9] Department of Virus and Microbiological Special Diagnostics, Statens Serum Institut, 2300 Copenhagen, Denmark
* Correspondence: ekaidoo@atu.edu.gh

**Abstract:** Coronavirus disease 2019 (COVID-19) contact tracing and malaria reactive case detection (RACD) are effective strategies for disease control. The emergence of the COVID-19 pandemic and the global attention COVID-19 has received in the recent past and present has hampered malaria control efforts. Among these are difficulties in finding and treating malaria-infected individuals in hypoendemic settings in the community, due to lockdown restrictions by countries. It is common knowledge that malaria cases that cannot be identified remain untreated. To sustain the gains made in malaria control, we proposed a two-pronged hybrid approach for COVID-19 contact tracing and malaria RACD in communities with COVID-19 and malaria coinfections. Such an approach would equally factor the burden of malaria cases and COVID-19 to support an effective strategy for responding to current and future pandemics.

**Keywords:** COVID-19; malaria; contact tracing; reactive case detection; strategy; hypoendemic settings; pandemic

## 1. Introduction

Contact tracing is an acceptable strategy of infectious disease control, mainly for illnesses involving direct transmission from person to person, such as COVID-19 [1]. It involves reaching out to patients with confirmed infection and working with them to recall everyone with whom they have had close contact during the period in which they may have been infectious [2]. The end goal is to identify the maximum number of COVID-19-infected persons and contacts in order to stop transmission through isolation and intervention [2]. There were 176 million confirmed cases of COVID-19 and 3.8 million deaths reported by the mid-year of 2021 [3]. The highest prevalence was in the American, Southeast Asian and European regions [3,4]. However, Africa and the Western Pacific World Health

Organization (WHO) Region recorded comparatively low numbers of cases and deaths [3,4]. In the African region majority of the cases and deaths were from the northern and southern parts, with case counts of 3.6 million and 89,000 deaths [5,6].

Reactive case detection (RACD) for malaria, on the other hand, employs information on passively detected malaria index cases in order to target potential secondary cases [7] in the community. In RACD, teams respond to cases by testing and treating household occupants and neighbors of an index case [8]. RACD aims to seek individuals with asymptomatic infection in order to provide treatment to interrupt onward disease and vector transmission [9] without necessarily having to test or treat the entire population. About 95% of the global malaria burden is reported in the WHO African region [10]. In 2020, following the COVID-19 pandemic, the estimated number of malaria cases increased to 241 million cases, an upsurge of 14 million cases relative to 2019 [10]. In that same year, malaria deaths rose to about 627,000, a 12% increase from 2019 [10]. According to the WHO, worldwide, 29 out of the 85 malaria-endemic countries were responsible for about 96% of malaria cases and mortalities in 2020 [10]. Nigeria (26.8%), Democratic Republic of Congo (12.0%), Uganda (5.4%), Mozambique (4.2%), Angola (3.4%) and Burkina Faso (3.4%) made up 55% of all cases [10]. Worldwide, four countries were responsible for just over half of all malaria mortalities. These were Nigeria (31.9%), the Democratic Republic of the Congo (13.2%), the United Republic of Tanzania (4.1%) and Mozambique (3.8%) [10].

The emergence of COVID-19 has a toll on malaria elimination efforts [11]. The advent of COVID-19 has left in its wake a disturbance in the continuity of global malaria interventions, such as seasonal malaria chemoprevention (SMC) and insecticide-treated bed nets (ITNs) distribution [11]. This is because of the global attention received by COVID-19 and restrictions employed, though malaria and COVID-19 present with common symptoms such as fever, breathing difficulties, tiredness and acute onset headache [11]. COVID-19 could be mild to severe, and 2% of the diagnosed die from acute respiratory distress syndrome (ARDS), while acute lung injury or ARDS is the main symptom of severe malaria. This can occur even after antimalarial treatment [12,13]. Malaria prevention and treatment in hypoendemic areas is even more crucial during the era of the COVID-19 pandemic to sustain the gains made [14]. In the last two decades, gains have been made in the mass distribution of long-lasting insecticidal nets (LLINs), seasonal malaria chemoprevention (SMC) and the indoor residual spraying of insecticide (IRS) [15].

However, in some areas, COVID-19 protocols made it impossible for health workers and trained persons to reach out to individuals with fever, potentially compromising progress towards malaria elimination [16]. Lack of infection-control prevention knowledge and the unavailability of personal protective equipment by community and primary healthcare workers activities makes it difficult to handle safely suspected COVID-19 cases [16]. It is uncertain how long the COVID-19 pandemic will last and how often similar pandemics will happen in the future; therefore, what hitherto used to be effective (RACD for malaria) must be strengthened in the era of COVID-19 to cater for patients with malaria and prepare for future disease outbreaks. According to the WHO, in most malaria endemic countries, the total number of the population fully vaccinated against COVID-19 is below 5% [10]. In these malarious countries, in excess of 101 million COVID-19 cases and 2.4 million deaths were reported [10]. Approximately, 47,000 (68%) of the extra 69,000 malaria deaths were a result of COVID-19 pandemic disruptions [10]. Malaria and COVID-19 present similar symptoms and geographic areas of disease spread. However, in pandemic situations, such as COVID-19, we run the risk of neglecting other known diseases, such as malaria. The burden of avoidable morbidity and mortality from such known diseases as malaria during pandemics ultimately leaves significant casualites of the very pandemic that has occasioned the neglect [17]. Approximately, 47,000 (68%) of the extra 69,000 malaria deaths were a result of COVID-19 pandemic disruptions [10].

In light of the above, the importance of COVID-19 contact tracing and the RACD of malaria to disease control cannot be overemphasized. Despite the WHO surveillance benchmarks of 2012, for malaria elimination in endemic regions, and the Centers for

Disease Control and Prevention (CDC)-published guide on contact tracing and operational knowhow by countries, adequate suggestions on how to execute both strategies across epidemiological areas are not available [18,19].

To ensure that malaria control efforts are sustained and patients with malaria identified, a need exists for practical examples of combining effective COVID-19 contact-tracing operations and malaria RACD on a large scale within countries. Sharing these strategies and adapting them at the local level is important as countries restart their economies, reopen institutions such as schools and continue to dispatch teams of contact tracers for COVID-19 testing and tracing [20].

In this article, we propose how COVID-19 contact tracing and the RACD of malaria could be achieved simultaneously in resource-limited settings with their attendant commonalities, differences and limitations in literature. We propose that, in the larger framework of contact tracing, RACD can be integrated in order to maximize malaria case detection and prompt treatment. The scope and reach of contact tracing invariably extends far and beyond RACD. Hence, taking RACD as a subset of contact tracing, both strategies can successfully be executed.

## 2. Key Definitions

**Contact** is any person exposed to an index case within the past 14 days. Exposure may be direct or indirect (casual), identified or unidentified. Direct exposure (occupying a living, working area, etc.) is easily identified, while casual exposure (contact on public transport, social gatherings, etc.) may be unidentified [21].

**Household member** is a person who shares the same enclosed living area with the index case. Household is living in the same compound but not in the same enclosed space with the index case [21].

**Index case** is the case around which an investigation revolves. The investigation normally centers on a defined individual or group of potentially exposed persons in which other (secondary) cases may be found. Hence, the index case is normally the case identified at the onset, although she or he may not be the source case [21].

## 3. Two-Pronged Hybrid Proposal for COVID-19 Contact Tracing and Malaria RACD

During COVID-19, the possibility of testing people who are negative, but present with fever, cannot be ignored at the expense of malaria [11]. Conversely, such people could be tested for malaria even when COVID-19 is the primary suspicion [11]. Furthermore, a patient may have COVID-19 and malaria co-infection and the diagnosis and treatment of one of them may lead to missing the other [11]. Given the above scenarios, to ensure continued access to diagnosis and care for patients with malaria during a COVID-19 pandemic, a two-pronged hybrid approach for COVID-19 contact tracing and malaria RACD is being proposed. COVID-19 contact tracing is normally triggered by a confirmed index case identified through symptom-based surveillance. It consists of three main steps: identifying, listing and monitoring persons who have been exposed to infected individuals, with the aim of quickly diagnosing, and preventing further spread of, infection [22]. Contacts of the index case are identified through interviews by public health officials (manual contact tracing) or tracking proximity records on digital devices (digital contact tracing) and asked to quarantine in order to prevent further transmissions.

In this proposal, we suggest that, following a case identification for COVID-19, contact identification and listing should be undertaken by case-investigation teams. This should involve interviewing the index case and/or family members to elicit an initial list of potentially exposed persons. Thereafter, contact tracers from the community locate the listed contacts and identify any additional contacts excluded from the initial investigation. Gathered information (location, unique case identifier for which contacts were listed, etc.) should then be recorded. Within this broader scheme detailed above, granted that infection control and prevention measures are implemented, malaria RACD should be able to be

integrated with well-established COVID-19 contact-tracing strategies and networks for community case finding.

When an individual has malaria and COVID-19 coinfection, at the time of sampling contacts (restricted to persons who have had direct exposure to the index case in the household), blood samples can be taken for all members of the household present (direct or indirect exposure with index case), as well for malaria screening. Regarding neighbors (non-household members), COVID-19 contact tracing could only be extended to those who had direct contact with the index case (Figure 1). According to a United Kingdom survey, the manual contact tracing of non-house-hold members would reduce incidence by 5–15%, coupled with the effect of quarantining symptomatic individuals and their household members [23]. Although it sounds logical, sometimes in the poorer households, quarantining symptomatic household could be challenging for the families. For example, they could be the only breadwinners of the family. During quarantine, such households are likely to suffer job losses, for which reason employers should be encouraged to safeguard the jobs of employees who require quarantine or treatment.

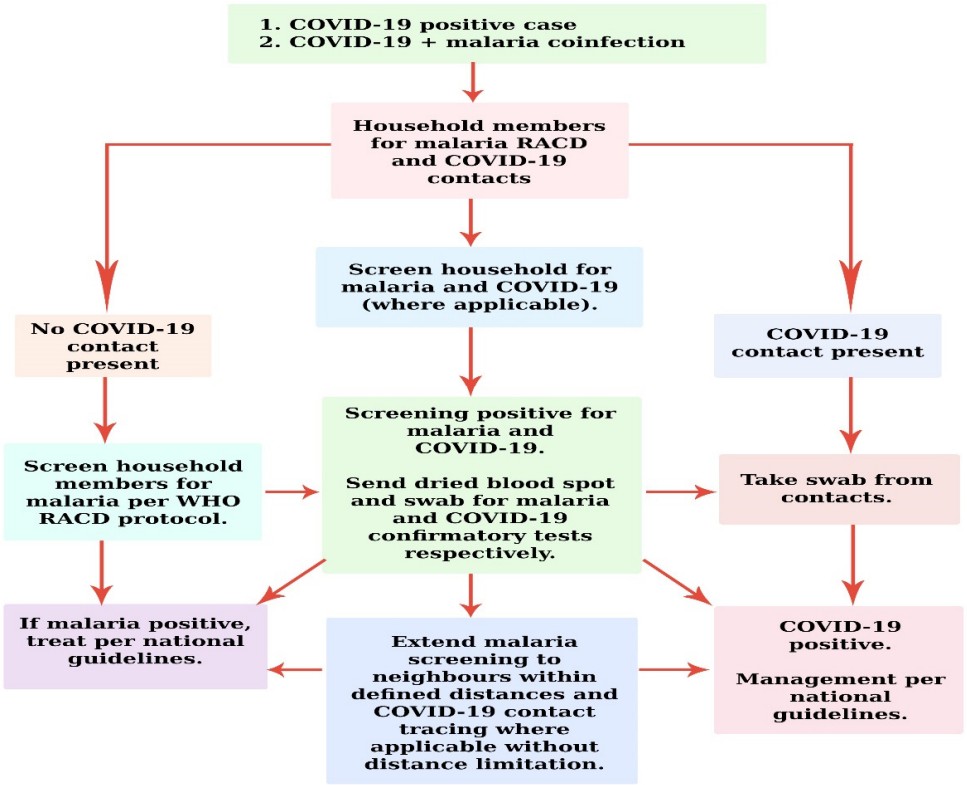

**Figure 1.** Schematic proposal for two-pronged hybrid COVID-19 contact tracing and malaria RACD: "adapted with modification and permission from [24]. 2021, USAID".

Detecting malaria, RACD can be informed by screening individuals (direct or indirect exposure with index case) residing within a particular distance of the index case or choosing a definite number of neighbors or households for follow-up [25] (Figure 1). The World Health Organization surveillance recommends reaching a sizeable population, with the view that the flight distance of *Anopheles* is usually 1–2 km [9].

## 4. Operational Commonalities between COVID-19 Contact Tracing and Malaria RACD

Both COVID-19 contact tracing and malaria RACD are triggered by the passive surveillance of index cases. The identification of COVID-19 patient begins with the symptomatic screening of high-risk patients such as healthcare workers or patients with a history of

contact with a confirmed COVID-19 case [11]. Similarly, malaria RACD starts with the diagnosis of a symptomatic malaria patient at a health facility [7]. Index cases involved in both disease conditions represent the foci of infection and hence the need for follow up. Such follow ups are undertaken with the intention of identifying additional secondary cases that are common to both COVID-19 contact tracing is undertaken by engaging identified cases, to further identify individuals who have been in contact with those identified cases [26].

Furthermore, COVID-19 contact tracing and malaria RACD as targeted measures aim to prevent the spread of further transmission where and when they occur. This can be fully effective on condition that all infectious individuals are identified (through symptoms or testing) before they infect others. Asymptomatic individuals identified through both strategies and tested might equally be infectious also, leading to the identification of asymptomatic hotspots such that places or risk groups that seed transmission to others may be identified and targeted.

Regarding COVID-19, a large proportion of transmissions occur prior to symptom manifestation [27,28]. The proportion of malaria-infected persons who are asymptomatic, who are minimally symptomatic or who do not seek medical care can be substantial and as high as 96% [29]. Malaria RACD is favorable to hypoendemic areas in the same way as contact tracing is normally most applicable during troughs of the epidemic curve, when such efforts are more manageable [30].

## 5. Differences

COVID-19 contact tracing involves the isolation of index cases and contacts who become infected during the infectious period of the disease [26,31] (Figure 2). The ability of this strategy in limiting COVID-19 spread hinges on rapid, extensive and precise execution [26]. Most of all, it is important to curtail delays in diagnostics, contact identification and the subsequent isolation of all possibly infected persons [32].

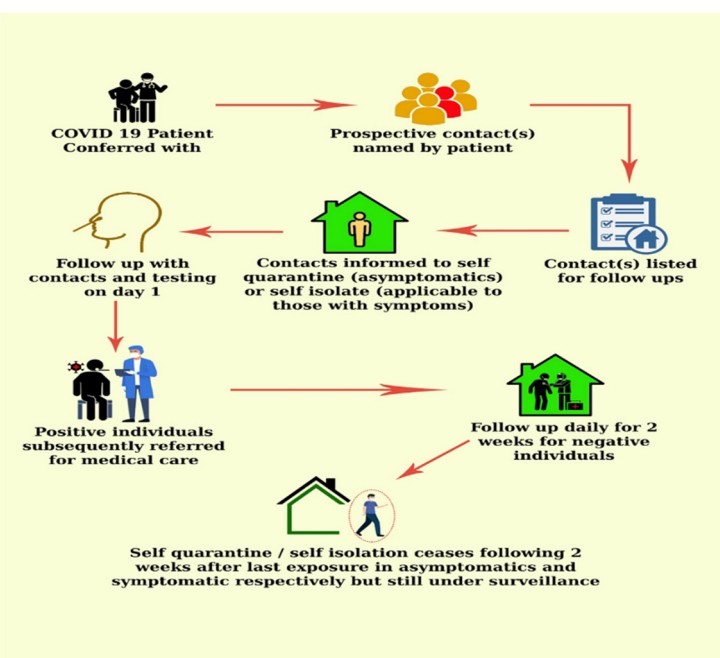

**Figure 2.** COVID-19 contact tracing flow chart.

It is, however, not so in malaria RACD, in that index cases and potential secondary cases are not asked to isolate. Malaria RACD does not discriminate. Rather, it focuses on all members of the index case household and neighbors within a certain radius who may not necessarily have come into contact with the case, but it is presumed that they might have come into contact with the vector (Figure 3). Larval hatching in *Anopheles* vector mainly takes place 2–3 days after oviposition based on environmental conditions in water, such as

oxygen availability and bacterial composition [33,34]. The complete pre-erythrocytic phase takes about 5–16 days, depending on the parasite species: usually 5–6 days for *Plasmodium falciparum*, 8 days for *P. vivax,* 8–9 days for *P. knowlesi,* 9 days for *P. ovale* and 13 days for *P. malariae* [35]. Merozoites are ultimately unleashed into the bloodstream in the lung capillaries and begin the blood stage of the infection afterwards [36]. The erythrocytic cycle takes place every 24 h in case of *P. knowlesi;* 48 h in cases of *P. falciparum, P. vivax* and *P. ovale;* and 72 h in case of *P. malariae* [37].

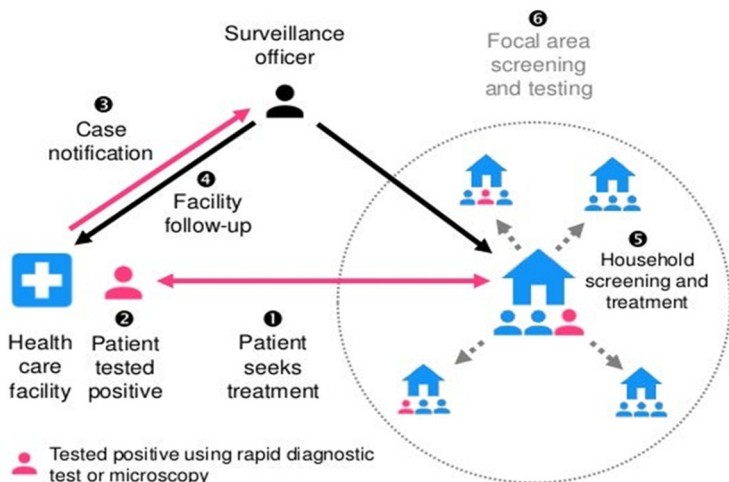

**Figure 3.** Reactive case detection of malaria:"reprinted /adapted with permission from [38] 2017, Gordon Cressman".

In malaria RACD, because the index case may be coming from the same household or neighborhood, enrollment is more likely, as subjects may be more willing to participate when the index case is known to them. Conversely, in COVID-19 contact tracing, infected and exposed individuals (contacts) may not necessarily be known to an index case, except when the contacts are within the same household or neighborhood as the index case. Hence, COVID-19 contact tracing is characterized by perceived stigma and mistrust, unlike the receptivity received by malaria RACD. In malaria RACD, more than 7 days can be taken to trigger the strategy [39]. This, however, may be unsuitable for COVID-19 contact tracing [40], because of its high transmissibility.

## 6. Challenges and Way Forward

A number of challenges are associated with these strategies. These include: resource limitations, improper coordination, communication difficulties, quarantine enforcement and the provision of isolation housing, among others. There is also improper public education about the coronavirus. Some people are unaware of the problems associated with the disease. Hotspots with no or low population access to healthcare may be missed. Moreover, the same can be said of index-case misclassification as false negatives, based solely on rapid antigen testing, to trigger COVID-19 contact tracing and malaria RACD or where the setting is considered unreceptive to undertaking these strategies. Such challenges can be resolved by the use of high-resolution risk maps based on case data to inform the classifications of areas by receptivity and to guide the implementation of such strategies in high-risk areas underserved by the health system. The use of highly sensitive molecular diagnostic tools with real-time results will come in handy for index-case misclassification as false negatives.

Furthermore, COVID-19 contact tracing and malaria RACD are slow, labor- and logistically intensive [31,41,42]. Index cases found within a particular setting at different times could require overlap in screening. Index-case household members and neighbors or contacts that were tested in a previous screening could be part of a subsequent index case, requiring repeated screening.

Frequent sampling and COVID-19 and malaria testing with few positive results could result in participation fatigue and possible refusal. By extension and fatigue-related, community-based health workers may see their efforts as unproductive, resulting in few positive outcomes and not needed in the event of unwillingness to participate [43]. In Nigeria and Rwanda, mistrust was addressed by involving both public and private authorities in sensitization campaigns to minimize the impact [44].

Additionally, locating contact-persons and index-case households can be impeded by challenges such as persons with no address, locations without street names, inaccessible roads and telecommunication unavailability, among others. These can be addressed by the use of locally known landmarks, the sole reliance on physical descriptions of cases in the absence of names and the provision of transportation and mobile phones for personnel. Furthermore, there is perceived stigma (shame, discrimination) associated with being a contact person (from peers, family, and/or the community). This has occasionally led to contact persons fleeing from follow-up and to non-adherence to self-isolation and quarantine. The promotion of positive health messaging campaigns and monitoring adherence to self-isolation and quarantine is one of the ways to mitigate the effect [44].

Relative to COVID-19 contact tracing, the challenges include the ability of individuals to recall and identify their contacts [40]. In household settings, though remembering contacts may be feasible, the ability to remember and identify close-range proximity contacts is challenging. Retrospective studies have shown that brief contacts have a lower probability of being recalled and that contact durations are overestimated [45]. Moreover, in settings such as public transportation, shops or elevators, sharing the proximity of unknown persons is a common feature. Smartphone technology (e.g., bluetooth) could be employed to identify and notify contacts whose phones are running the app when at risk of infection [40,46]. Exposed individuals warned via digital contact tracing (DCT) are asked to contact the responsible health authorities and quarantine. The efficiency of contact-tracing apps is obviously limited by their numbers of users, since both the infectious person and their contacts need to have installed the app, and by the compliance of alerted users. There is also the issue of data privacy concerns, among other controversies [47,48].

## 7. Conclusions

Both COVID-19 contact tracing and malaria RACD are effective public health strategies for disease control. To ensure that malaria-control efforts are sustained in the wake of the challenges imposed by the COVID-19 pandemic or any future pandemic, the community-level case identification of malaria should be equally prioritized. Strategizing and simultaneously dealing with COVID-19 and malaria case finding would help optimize resource requirements and enhance service-delivery efficiency for both diseases.

## 8. Patents

The authors thank Gordon Cressman and the USAID for permission to use copyrighted images on the reactive case detection of malaria, the adapting and modifying of household-level tuberculosis contact investigation and COVID-19-screening algorithms.

**Author Contributions:** Conceptualization, writing—original draft preparation E.K.A.; software, visualization O.O.-D.A.; writing—review and editing D.S.S., K.B., F.A.B., G.O.-A., S.A.S., L.A., M.A., R.D.-T., R.H.A., B.W.L. and K.A.K. All authors have read and agreed to the published version of the manuscript.

**Funding:** Lundbeck Foundation (R349-2020-703) support research in COVID to Prof KA Krogfelt, RUC, DK.

**Institutional Review Board Statement:** Not applicable.

**Informed Consent Statement:** Not applicable.

**Data Availability Statement:** Not applicable.

**Conflicts of Interest:** The authors declare no conflict of interest.

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
