# Peer review of "Joint COVID-19 Contact Tracing and Malaria Reactive Case Detection as Efficient Strategies for Disease Control"

_covid, doi:10.3390/covid2090091_

Round 1

Reviewer 1 Report

The article “Combined COVID-19 contact tracing and malaria reactive case detection as efficient strategies for disease control: commonalities, differences and challenges” is well written. There are some minor suggestions for the authors to improve the manuscript.

The emergence of COVID-19 has a toll on malaria elimination efforts

Please provide references.

This is because of the global attention received by COVID-19 and restrictions employed, though malaria and COVID-19 present with common symptoms such as fever, breathing difficulties, tiredness, and acute onset headache.

Please provide references.

Line (57-58) : Please mention that, Covid-19 could be mild to severe and 2% of the diagnosed die from ARDS while acute lung injury or ARDS is the main symptom of severe malaria. This can occur even after antimalarial treatment.

The authors should cite these papers

Identification and Development of Therapeutics for COVID-19, Rando et.al., mSystems, 2021,

Could Heme Oxygenase-1 Be a New Target for Therapeutic Intervention in Malaria-Associated Acute Lung Injury/Acute Respiratory Distress Syndrome? Pereira et.al., Front. Cell. Infect. Microbiol., 2018.

Malaria prevention and treatment in hypoendemic areas is even more crucial during the era of COVID-19 pandemic to sustain the gains made.

Please elaborate on this.

According to a United Kingdom  survey, manual contact tracing of non-house-hold members would reduce incidence by  5–15%, coupled with the effect of quarantining symptomatic individuals and their house  hold members. [14].

Although it sounds logical, sometimes in the poorer household quarantining symptomatic household could be challenging for the families. For example, they could be only bread earners of the family. Please mention this in the paper. A better thoughtful management of the situation would have been useful.

A number of challenges are associated with these strategies. These include: resource limitations, improper coordination, communication difficulties, quarantine enforcement, provision of isolation housing amongst others

Please also mention the improper public education about the virus. Some people are unaware problems associated with the disease.

The use of highly sensitive molecular diagnostic tools with real time results will come in handy for index case misclassification as false negative.

Please mention about “Rapid antigen testing”.

Promotion of positive health messaging campaigns and monitoring adherence to self-isolation and quarantine is one of the ways to mitigate the effect.

Please mention the references.

Reviewer 2 Report

This article does not represent a research paper but rather an opinion article namely on how to deal with joint RACD/contact tracing around cases that have been identified as either a malaria or a covid case (or both) through passive surveillance

How to combine this is interesting - but not fully clear - the key Fig 1 is confusing : It is not clear as to what is the starting index case and what to do next . It should also be noted that an index malaria case is most often a child and an index Covid an adult - will that affect the algorithm ?

Reviewer 3 Report

TITLE

replace "combined" with "joint"
Keep either of the following phrases in the title"....as efficient strategies for disease control" AND ":commonalities, differences and challenges"

GENERAL
It would be better if the basic knowledge from epidemiology would be reflected in the strategy in order to ensure targeted approach and to limit cost i.e. time, place and person in terms of RACD. Although there was a clear-cut mention of the persons to be targeted but no mention of time and place: the time period focused would be the time vector hatches and parasites replicates more; the places focused would be malaria epidemic/endemic areas....

The reliable statistics on COVID-19 and malaria co-existence and/or ignoring either of these globally as well as regionally were not mentioned at all.

Why is it (area of current perspective) a issue of focus has not been debated, it should be based on the relevant statistics

In many places there was repetition of the information which needs to be removed as it makes this perspective unnecessarily long and needs to be shortened 

Round 2

Reviewer 2 Report

I see no change in Fig 1 - so to me the figure remains confusing

Reviewer 3 Report

Authors have incorporated all the suggested changes. This manuscript is in much improved form to be published. All the best

Author Response

Thanks very much for improving on our manuscript with your insightful comments.